# Navigation Algorithm Based on the Boundary Line of Tillage Soil Combined with Guided Filtering and Improved Anti-Noise Morphology

**DOI:** 10.3390/s19183918

**Published:** 2019-09-11

**Authors:** Wei Lu, Mengjie Zeng, Ling Wang, Hui Luo, Subrata Mukherjee, Xuhui Huang, Yiming Deng

**Affiliations:** 1College of Engineering, Nanjing Agricultural University, Nanjing 210031, China; zmj_njaurobot@163.com (M.Z.); Lingw@njau.edu.cn (L.W.); lh821005@njau.edu.cn (H.L.); 2Key Laboratory of Intelligent Agricultural Equipment in Jiangsu Province, Nanjing Agricultural University, Nanjing 210031, China; 3NDE Laboratory, College of Engineering, Michigan State University, East Lansing, MI 48824, USA; subratamukherjee@sina.com (S.M.); xuhui_huang@sohu.com (X.H.); dengyimi@egr.msu.edu (Y.D.)

**Keywords:** intelligent tractor, vision navigation, improved anti-noise morphology, boundary line, guided filtering

## Abstract

An improved anti-noise morphology vision navigation algorithm is proposed for intelligent tractor tillage in a complex agricultural field environment. At first, the two key steps of guided filtering and improved anti-noise morphology navigation line extraction were addressed in detail. Then, the experiments were carried out in order to verify the effectiveness and advancement of the presented algorithm. Finally, the optimal template and its application condition were studied for improving the image-processing speed. The comparison experiment results show that the YCbCr color space has minimum time consumption of 0.094 s in comparison with HSV, HIS, and 2R-G-B color spaces. The guided filtering method can effectively distinguish the boundary between the tillage soil compared to other competing vanilla methods such as Tarel, multi-scale retinex, wavelet-based retinex, and homomorphic filtering in spite of having the fastest processing speed of 0.113 s. The extracted soil boundary line of the improved anti-noise morphology algorithm has the best precision and speed compared to other operators such as Sobel, Roberts, Prewitt, and Log. After comparing different sizes of image templates, the optimal template with the size of 140 × 260 pixels could achieve high-precision vision navigation while the course deviation angle was not more than 7.5°. The maximum tractor speed of the optimal template and global template were 51.41 km/h and 27.47 km/h, respectively, which can meet the real-time vision navigation requirement of the smart tractor tillage operation in the field. The experimental vision navigation results demonstrated the feasibility of the autonomous vision navigation for tractor tillage operation in the field using the tillage soil boundary line extracted by the proposed improved anti-noise morphology algorithm, which has broad application prospect.

## 1. Introduction

Agricultural machinery [1] automatic navigation systems are a key part of smart tractors [2] for implementing precision agriculture, which can free drivers from boring work, as well as improve the work quality. A global positioning technique using GPS or GNSS [3,4,5] is applied for the automatic navigation of unmanned tractors during tillage operation in the field to obtain absolute geographic coordinates dynamically, whereas the navigation precision is heavily influenced by the inclination angle of the field surface, meteorological conditions, and the strength of the satellite navigation signal, especially in remote areas. Inertial sensors [6,7], which have a large error in long-distance navigation, can only be used as compensation for short-range position correction. Therefore, vision navigation [8], as a popular method in autonomous vehicles, mobile robots, and aircraft, was introduced to intelligent agricultural machinery. The Silsoe Research Institute of the United Kingdom focused on machine vision-based automatic navigation technology and established an extended Kalman filter model for field vehicles. The research team in Cemagref University in France proposed the MRF algorithm to deal with the edge recognition problem of harvested crops to identify crop rows from a multi-feature perspective. Nishiwaki of Japan used a pattern-matching method according to the characteristics of the distribution shape between rice rows to measure vehicle position. Carnegie Mellon University combined a visual odometry system with an aided inertial navigation filter to produce a robust tractor navigation system where the accuracy is measured in meters on rural and urban roads and, thus, does not rely on external infrastructure. Aiming at crop harvest and management operation, China Agricultural University, Nanjing Agricultural University, and Nanjing Forestry University proposed plant navigation line extraction algorithms such as filtering methods based on image scanning, wavelet transform, and optimized Hough transform, respectively, to improve recognition of plant navigation lines. 

A local autonomous navigation method by applying the boundary line of tillage soil is presented in this paper, with the tractor works in tillage mode inspired by the existing plant line navigation method. Moreover, a fast image-processing algorithm using optimized templates combined with guided filtering [9,10,11,12], improved anti-noise morphology, and Hough transformation [13,14] is proposed to meet the practical application of agricultural tillage.

## 2. Materials and Methods

Here, a traditional tractor of type AXION 850 (CLAAS) was modified into an intelligent tractor with the aid of a tractor-driving robot that drives the tractor as shown in Figure 1a. This tractor-driving robot consisted of a steering arm, a gear-shifting arm, a brake leg, a clutch leg, and an accelerator leg; it can operate a tractor imitating a tractor driver. The steering arm uses the motor to drive the steering wheel of the tractor through gears and chains, as shown in Figure 1b,c. Moreover, a camera (BLUELOVER, resolution 1280 × 980) and an RTK GPS (X10, Huace Co., Shanghai, China) were installed on the tractor for automated guided operation.

The tillage procedure of the intelligent tractor included two steps. Firstly, the tractor tilled a round line under manual control or teleoperation control to create a boundary between the tillage soil. Then, the tractor worked an autonomous navigation model based on the boundary calculated using guided filtering, improved anti-noise morphology, and Hough transformation in sequence, in consideration of the severe randomness and non-uniformity of the agricultural filed environment.

### 2.1. Guided Filtering

#### 2.1.1. Local Linear Model

The local linear model is mostly used for non-analytic functions, defined as the adjacent points on a function which have a linear relationship with others. The definition shows that a complex function can be represented by multiple adjacent local simple linear functions, where each point value can be obtained by averaging the weights of all the linear functions excluding that point.

The input image is denoted as I, which is the image to be filtered, while the output is denoted as q. The local linear model of the guided filter assumes that q is a linear transform of I in a window $w_{k}$ centered at pixel k; thus, $q_{i}$ which is a pixel in q can be expressed as
(1)$$q_{i}=a_{k}I_{i}+b_{k}\;,\;\forall i\in w_{k},$$
where a and b are the coefficients of the linear function when the window is centered at the pixel index value k, and i and k are the indexes of a pixel.

After a gradient operation on both sides of Equation (1), we get
(2)∇q=a∇I,
where q and I have similar gradients; thus, the output q maintains the same edge characteristics with I.

#### 2.1.2. Local Linear Model Solution

The process of calculating linear function coefficients is called linear regression. The true value of the fitting function p is defined, and the difference value between p and the actual output is as shown below.
(3)$$E\left(a_{k},b_{k}\right)=\sum_{i\in w_{k}}(\left(a_{k}I_{i}+b_{k}-p_{i}{)}^{2}+\varepsilon a_{k}^{2}\right),$$
where p denotes the image to be filtered, and ε is a parameter used for adjusting the filtering effect, whose purpose is to prevent a value from being too large. The filtering effect improves remarkably with the increment of ε. Minimizing Equation (3) transforms it to a least square problem within a certain window $w_{k}$, where the solution is given by
(4)$$a_{k}=\frac{\frac{1}{\left|w\right|}\sum_{i\in w_{k}}I_{i}p_{i}-\mu_{k}\bar{p_{k}}}{\sigma_{k}{}^{2}+\varepsilon },$$
(5)$$b_{k}=p_{k}-a_{k}\mu_{k},$$
where $\sigma_{k}{}^{2}$ and $\mu_{k}$ are the variance and mean of I in $w_{k}$, $\bar{p_{k}}$ is the average value of the image p to be filtered in the window, and |w| denotes the number of pixels contained in the window.

In view of the fact that a pixel can be described by multiple linear functions, all linear function values containing the point are weighted and averaged when calculating the output value of the point, as shown in Equation (6).
(6)$$q_{i}=\frac{1}{\left|w\right|}\sum_{i\in w_{k}}\left(a_{k}I_{i}+b_{k}\right)=\bar{a_{\tau }}I_{i}+\bar{b_{\tau }}.$$

The process of weighted averaging is a linear translational variation filtering process, and the calculation of the guided filtering is based on this process.

The processing process of the guided filter is shown in Figure 2. Here, the definition of a guiding graph is P, and the relationship between P and the original graph I is represented by a local linear model definition. In the figure, q is the linear transformation of the original image I in the window adjacent to a pixel value, and a and b are the coefficients of the linear function of the window center at that pixel value. The output pixel value q is linearly multiplied by the input image I; thus, q has a similar gradient to I. The edge characteristics of the original image I are still preserved in the image q after being processed by the guided filtering process.

### 2.2. Improved Anti-Noise Morphology Algorithm for Image Navigation Line Extraction

Introducing the concept of mathematical morphology [15,16,17,18,19] to the image edge detection operator can overcome the shortcomings of the classical operator [20,21,22] and can greatly reduce the calculation amount. This paper proposes an improved anti-noise morphology algorithm for image navigation line extraction, which selects a pair of smaller-scale structuring elements for further anti-noise processing to extract an image navigation line based on the edge feature.

In the mathematical morphology algorithm, structuring elements should be selected according to actual needs. Small-scale structuring elements can make the extracted image edges more detailed and coherent, and obtain more accurate edge localization, whereas large-scale structuring elements can reflect the large edge contours in the image and have a good noise suppression effect. Therefore, small-scale structuring elements were selected for obtaining complete edges in this paper.

The edge of the image is calculated as follows:(7)$$y_{d}=\left(\left(\left(f\circ A_{1}\right)\cdot A_{2}\right)⊕A_{2}\right)\circ A_{2}-\left(f\circ A_{1}\right)\cdot A_{2},$$
(8)$$y_{e}=\left(f\circ A_{1}\right)\cdot A_{2}-\left(\left(\left(f\circ A_{1}\right)\cdot A_{2}\right)\Theta A_{2}\right)\cdot A_{2},$$
where $A_{2}$ and $A_{2}$ are two different structuring elements, shown below.
$$A_{1}=\left[0,1,0;1,1,1;0,1,0\right],$$
$$A_{2}=\left[1,0,0;1,0,0;1,0,0\right]$$

In Equations (7) and (8), f is the image after guided filtering treatment. The anti-noise morphology edge detection operator is given as follows:(9)$$y_{de}=y_{d}+y_{e}.$$

Via some ordinary operations, the edge information can be easily obtained from the images of the edges detected by Equations (7) and (8). For detecting more detailed edges, as well as improving the anti-noise ability of $y_{de}$ under the condition of equal noise, the noise immunity is defined as
(10)$$y=y_{de}+E_{min},$$
where $E_{min}$ is $\min\{y_{d},y_{e}\}$, $y_{d}$ is the edge detected by Equation (7), $y_{e}$ is the edge detected by Equation (8), and $y_{de}$ is the edge detected by Equation (9).

The tillage soil boundary line extracted by using the algorithm mentioned above is shown in Figure 3b,c shows the boundary line extracted by color space conversion followed by threshold processing. There are remarkable errors at both ends of the navigation line in Figure 3b because of the calculation error caused by the truncation of the image, whereas it is obvious that the truncation error in Figure 3c is significantly improved for navigation line extraction.

The tractor completed the first returning tillage manually via human driving or tele-operational driving before implementing the autonomous image-aided navigation operation. The navigation line of the tractor was calculated by using a Hough transformation from the processed tillage soil boundary in Figure 3. In the actual operation, the navigation line was attached to one side of the tractor. It was influenced by the angle of view of the camera position, which resulted in an angular deviation between the calculated navigation line and the actual line. For this problem, the transverse line of the tractor was defined as the horizontal line $l_{h}$ The forward direction line of the tractor deviated in the camera image as shown in Figure 4, where the front lines $l_{f}$ in the left view and right view are rotated to an acute angle ($\theta_{0l}$) and an obtuse angle ($\theta_{0r}$) to the horizontal line $l_{h}$.

For simplifying the adjustment algorithm of the navigation line, the actual direction angle θ of the tractor was calculated as follows:(11)$$\theta =\varphi \cdot k\;(k=\frac{90{}^{\circ}}{\theta_{0}}),$$
where φ denotes the navigation angle between the tillage soil boundary and the horizontal line of the tractor extracted from the image, $\theta_{e}{}^{\prime }$ is the angle between the front line and the tillage soil boundary, and $\theta_{0}$ is the angle between the front line and the horizontal line in the image, equal to $\theta_{0l}$ and $\theta_{0r},$ respectively, when the boundary line gets located at the left side and right side of the tractor.

During tractor navigation using the tillage soil boundary line, $\theta_{0}$ and k are calculated as initialization. Then, the navigation angle θ is obtained after navigation line extraction from the image. 

The tractor needs to turn left if the navigation angle θ is not more than 90°; otherwise, it needs to turn right whether the boundary line is located on the left or right side of the tractor. The steering adjustment algorithm flowchart is shown in Figure 5. 

The detailed algorithm of improved anti-noise morphology is shown in Figure 6.

## 3. Experiment

In this paper, the computer employed for the tillage soil boundary navigation line extraction was configured as follows: 64-bit Windows 10 operating system, 8 GB memory, and a two-core processor Intel (R) Core (TM) i5-4200H central processing unit (CPU) @ 2.80 GHz. The first experiment was carried out to verify the effectiveness of the proposed algorithm mentioned above using several tillage images of different farms. Secondly, the navigation algorithm was used for smart tractor tillage in an experimental farm of Nanjing Agricultural University. Moreover, the optimal template was studied to improve the efficiency of the algorithm.

### 3.1. Effectiveness Verification of the Algorithm

#### 3.1.1. Color Space Selection

The time consumption and each optimal effect component of the YCbCr, HSV, HIS, and 2R-G-B [23,24,25,26] format images converted from the original image are shown in Table 1 and Figure 7, which indicate that the tillage soil boundary in the Cr component in the YCbCr format image was clearest among all the gradation component images, as well as having the fastest conversion speed of around 0.094 s. 

Furthermore, the histogram of the best gradation component in each image, shown in Figure 8, denotes that the frequencies of I and V gradation components were constantly zero; the 2R-G-B gradation component had a less obvious threshold segmentation pixel number, whereas the Cr gradation component had an apparent threshold segmentation point at 15 pixels. Thus, the Cr component was selected for the identification of the tillage soil boundary.

#### 3.1.2. Filtering Method Selection

As the illumination strength in the field does not remain constant, to make the algorithm robust, images used in filter selection were captured in both weak and strong light environments, as shown in Figure 9.

The homomorphic filtering [27,28,29,30] method is very popular in image enhancement processing because it can enhance the abrupt component (demarcation line) and suppress the slow variation component at the same time. The two images with weak light intensity and strong light intensity were processed using the homomorphic filtering method. Processes and the corresponding results under both the weak and strong light intensities are shown in Figure 10 and Figure 11, respectively. However, as shown in the below figures, the homomorphic filtering method was not suitable for extracting the boundary line between tillage soil as the slow varying components were more in proportion than the abrupt components in the field tillage images.

The homomorphic filtering (HF) method and other algorithms such as Tarel [31], multi-scale retinex [32,33,34], wavelet-based retinex [35], and guided filtering method were applied to enhance the boundary distinction between tillage soil for contrast testing, as shown in Figure 12. The Tarel algorithm could clarify the soil boundary but intensified the dry straw simultaneously, which led to huge image interference. In contrast, the multi-scale retinex algorithm could eliminate the dry straw information but weakened the reflection difference between the tillage soil, which resulted in low discrimination of the soil boundary. Both the wavelet-based retinex algorithm and homomorphic filtering algorithm could enhance the soil boundaries but darkened or brightened the whole images, which made it difficult for them to identify the boundaries using the binarization method. The guided filtering algorithm could enhance the contrast between the tillage soil, which made it convenient for the boundary extraction.

The contrastive calculation time of the filtering algorithms mentioned above is shown in Table 2. Among them, the image processing time of the guided filtering algorithm was only 0.113 s, followed by the multi-scale retinex, HF, Tarel, and wavelet-based retinex algorithms which took 0.552 s, 0.867 s, 0.902 s, and 1.008 s, respectively.

Therefore, the guided filtering algorithm was selected for tillage soil boundary identification, considering the image-processing speed and the contrast of the tillage soil.

#### 3.1.3. Navigation Line Extraction Using Improved Anti-Noise Morphology Algorithm

The edge extraction results and the Hough transform results of the image obtained using the basic morphology and improved anti-noise morphology algorithms are shown in Figure 13 and Figure 14, respectively. The improved anti-noise morphology algorithm could eliminate the boundary noise made by the basic morphology algorithm because of the adopted double structure and minimum operation. The navigation error decreased from 10° using the basic morphology algorithm to 0.5° using the improved anti-noise morphology algorithm.

Different edge extraction results obtained by applying the improved anti-noise morphology algorithm and other popular existing edge extraction algorithms such as Sobel [36,37,38,39,40], Roberts [41,42], Prewitt [43,44], and Log [45] are shown in Figure 15. There were some breakages in the extracted edges using the latter four algorithms, which led to large errors during navigation line identification processing. Moreover, the longest extracted navigation line created using the improved anti-noise morphology combined with Hough transform method had the best precision compared with the others as shown in Figure 16.

The time consumption of the above algorithms is shown in Table 3. Among them, the time consumption of the improved anti-noise morphology was minimal, followed by the Sobel operator, the Roberts operator, the Prewitt operator, and the Log operator. The longest line was used as the tractor navigation line to calculate the orientation error for steering adjustment of the tractor.

#### 3.1.4. Image Template Optimization

In order to further improve the real-time vision navigation during linear tillage operation of the tractor, the appropriate image should be cropped from the whole original image, because a larger picture needs more computer processing time. The tractor tillage operation includes two work modes, i.e., a linear mode and a turning mode. During the linear mode, the longest line of the tillage soil boundary is firstly created using the above improved anti-noise morphology algorithm. Then, a rectangle centered at the middle of the longest line is used to crop a part of the original picture for navigation line calculation, because the position and the navigation angle of boundary line of the soil change a little. During the turning mode, the original image should be adopted for navigation because the position of the boundary in the image varies a lot, which leads to boundary information loss in the rectangle.

The optimization template selection scheme is as follows:(1)The original image is transformed to grayscale and uniformly scaled to 816×612 pixels.(2)The middle of the longest line is used as a reference point.(3)Different size rectangles centered at the reference point are used for navigation precision comparation, as shown in Figure 17a.

The experimental results, shown in Figure 17b, denote that the navigation angle error decreased significantly when the length and width were not less than 260 and 140 pixels, respectively. When the template was less than 140×260 pixels, the boundary information decreased with the shrinkage in template size, as the size of the boundary signal in the image below that template size was not more than that of the background noise. Moreover, the processing of the algorithm took only 47 ms for the template size of 140×260 pixels compared with that containing 816×612 pixels, which had a time consumption of 520 ms.

### 3.2. Navigation Experiment in the Field

The autonomous navigation experiment using the proposed tillage soil boundary was carried out in the farmland of Nanjing Agricultural University using a tractor (CLAAS AXION 850 model) equipped with the tractor-driving robot developed by our team. For comparing the navigation precision, an RTK GPS (X10, Huace Co., China) was applied, and the navigation line was used as the reference, as shown in Figure 18.

Before the autonomous navigation tillage operation, the tractor was teleoperationally controlled for the first round trip of tillage, and then the tractor longitudinal direction was adjusted to be parallel to the tillage soil boundary line. A navigation line identification diagram in the experiment is shown in Figure 19.

Firstly, the applied condition on the optimal template containing 140×260 pixels was studied by altering the deviation angle $\theta_{e}$ between the tractor’s longitudinal direction and the boundary between the tillage soil. The experimental results (Figure 20) indicate that the vision navigational deviation error δ increased dramatically when $\theta_{e}$ was more than 7.5°, which means that the optimal template was suitable for the tractor body course error range of $\theta_{\mathrm{em}}=\left[-7.5{}^{\circ},\;+7.5{}^{\circ}\right]$. Then, the limit of the tillage operational speed of the tractor was studied based on the total processing time of the algorithm t=0.33 s, the maximum permissible deviation angle $\theta_{\mathrm{em}}$, the physical field size corresponding to the optimal template of 140×260 pixels, and the maximum deviation distance $d_{\max}$.
(12)$$d_{max}=\frac{0.62}{sin7.5^{{}^{\circ}}}=4.75m;$$
(13)$$v_{max}=\frac{d_{max}}{t}=14.40m/s=51.41km/h.$$

Therefore, the ideal maximum working speed of the tractor was 51.41 km/h for vision navigation using an optimized template when $\theta_{e}$ was 7.5°. The maximum working speed of the tractor increased sharply when $\theta_{e}$ decreased, as shown in Figure 21. Actually, the existing literature shows that the tractor speed is not more than 5.2km/h while tilling. 

Based on the above study, the vision navigation method is shown in a flowchart (Figure 22). The optimal template is adopted when the deviation angle $\theta_{e}$ is less than 7.5°; otherwise, the global template is applied for navigation.

## 4. Discussion and Conclusions

Automatic navigation in the field is a pivotal technology for intelligent agricultural machinery [46]. Satellite navigation technologies such as GPS and BDS are applied in field operations of tractors through the combined use of RTK technology with the disadvantage of high cost for reaching centimeter-level navigation accuracy; however, this absolute position navigation technology leads to a cumulative error. In addition, navigation failure or large errors also occur due to the complicated farmland environment, such as the uneven ground and micrometeorology of remote farmland, which generates signal interruptions [8]. LiDAR was used for agricultural robot navigation in orchards [47] and seedling crops [48]; however, it needs tall plants or trees as a reference. Thus, the visual navigation method was introduced to agricultural machinery navigation. The central position of a cotton seedling row was identified as a navigation line for tractor navigation according to the regional vertical cumulative distribution graph of the images [49]. Green maize seedlings were identified, and the accumulation of green pixels was used to extract curved and straight crop rows for tractor navigation [50]. Most previous studies extracted the visual navigation line based on the color difference between the crops and the background. 

Inspired by the existing visual navigation methods in the field, this paper put forward a visual navigation method based on the color difference between the tillage soil during tillage operation, and the proposed improved anti-noise morphology vision navigation algorithm combined with guided filtering algorithm worked well in a complex agricultural field environment with a non-uniform nature, uneven illumination, and straw disturbance. The accuracy of the navigation angle of the proposed algorithm was not less than 0.5°, compared with the precision of 1.15° using a combination of RTK–GPS and IMU [51].

The optimal size of the image was studied for improving the algorithm speed. The optimal template of 140×260 pixels is applied when the deviation angle $\theta_{e}$ is less than 7.5°. At this time, the navigation line extracted from the edge information processed by the improved anti-noise morphological operator with the aid of Hough transform was the most accurate and the fastest, with a time consumption of only 0.047 s. Under this optimal circumstance, the real-time navigation of the tractor body could be satisfied as long as the tractor speed was no more than 51.41 km/h. In our earlier discussion, the image-processing time of the guided filtering algorithm was only 0.113 s, followed by the multi-scale retinex, HF, Tarel, and wavelet-based retinex algorithms which took 0.552 s, 0.867 s, 0.902 s, and 1.008 s, respectively. Moreover, as shown earlier, the time consumption of the improved anti-noise morphology was minimal, followed by the Sobel operator, the Roberts operator, the Prewitt operator, and the Log operator. However, the global template is used to satisfy the real-time visual navigation of the vehicle body when the course deviation is greater than 7.5°. and the speed of the tractor is no more than 27.47 km/h.

The experiment results show that the navigation line extraction algorithm in this paper took less time and had a good effect in a complex farmland environment. Therefore, the fast navigation line extraction method based on improved anti-noise morphology has the advantages of short time consumption and high precision, and it can meet the requirements of real-time vision navigation in field tillage using an intelligent tractor, which has important practical application value.

Furthermore, a comparison between the existing operators and the proposed improved anti-noise morphology algorithm was carried out using standard datasets (downloaded from https://www.sensefly.com/education/datasets/); the results showed that, overall, the time consumption and navigation precision of our proposed algorithm have significant advantages compared to other previous operators, as shown in Figure 23.

Future work will introduce the thin-plate spline interpolation algorithm to calibrate colors in sRGB space [52], as well as a depth camera [53] and advanced artificial intelligence algorithms [54,55] to improve navigation precision, speed, and robustness during tractor tillage operation.

## Figures and Tables

**Figure 1 sensors-19-03918-f001:**
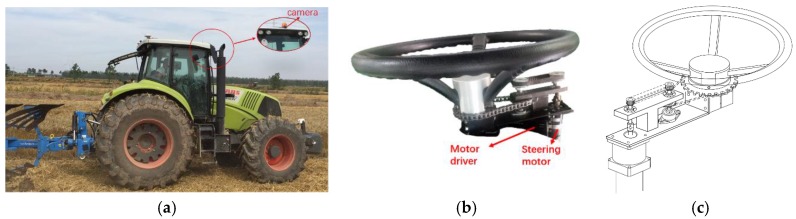
Intelligent tractor platform: (**a**) tractor platform; (**b**) steering configuration; (**c**) structural schematic diagram of the steering configuration.

**Figure 2 sensors-19-03918-f002:**
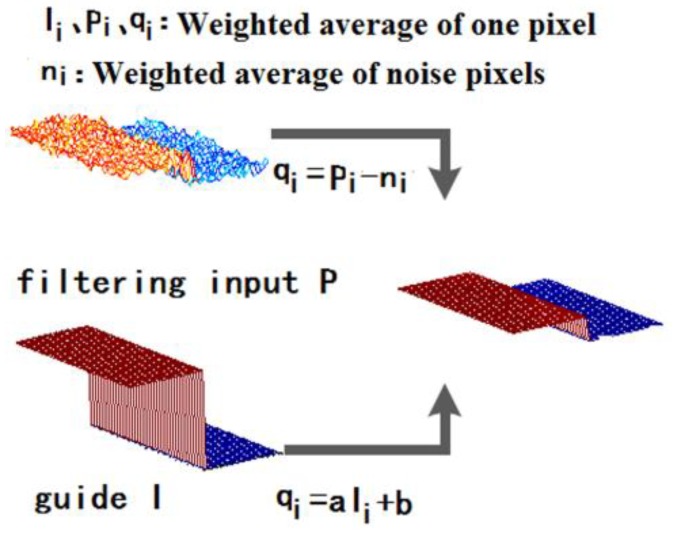
The guided filtering process.

**Figure 3 sensors-19-03918-f003:**
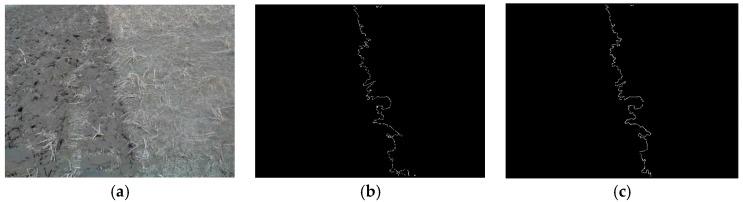
The edge processing comparison: (**a**) original; (**b**) edge information; (**c**) processed edge information.

**Figure 4 sensors-19-03918-f004:**
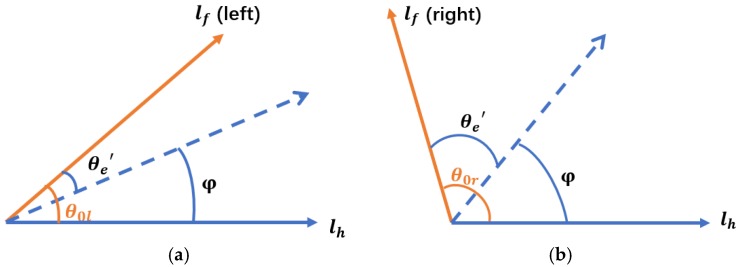
Direction correction diagram: (**a**) left view; (**b**) right view.

**Figure 5 sensors-19-03918-f005:**
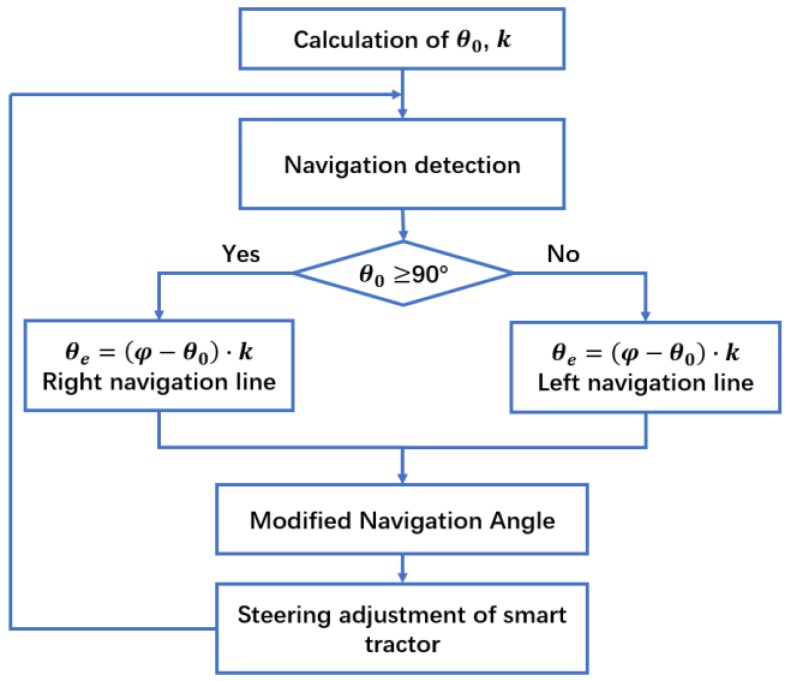
Steering adjustment algorithm flowchart.

**Figure 6 sensors-19-03918-f006:**
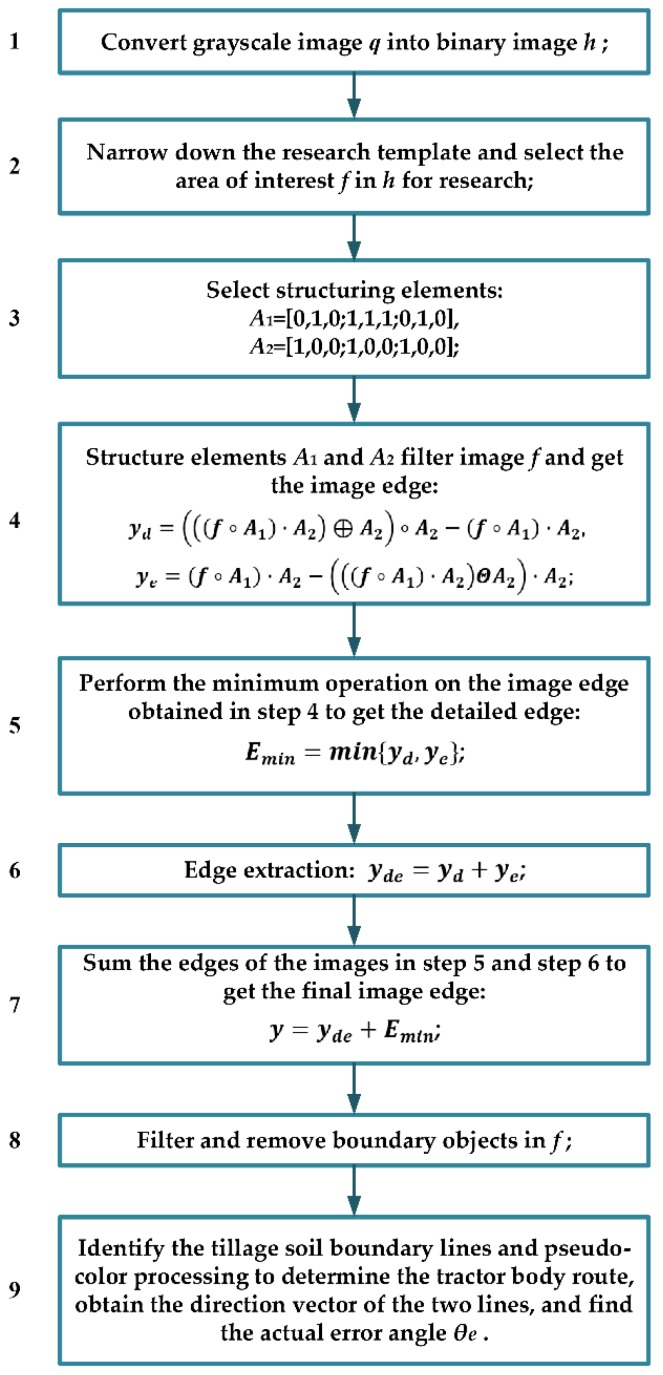
Improved anti-noise morphological algorithm.

**Figure 7 sensors-19-03918-f007:**
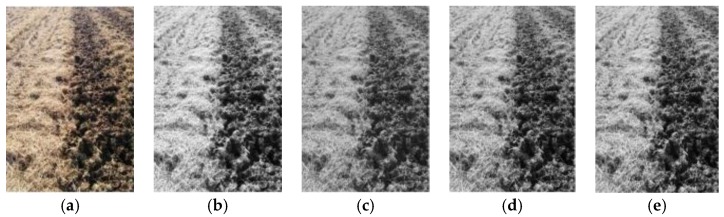
Component graph of each color space: (**a**) original; (**b**) Cr; (**c**) V; (**d**) I; (**e**) 2R-G-B.

**Figure 8 sensors-19-03918-f008:**
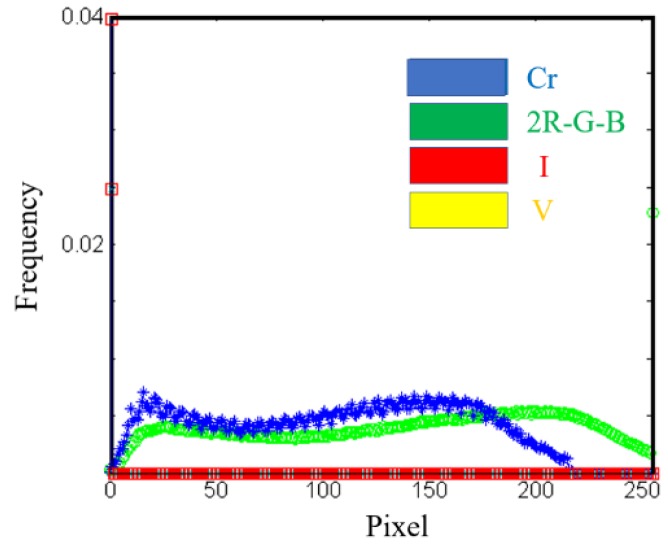
Histogram comparison of Y, V, I, and 2R-G-B.

**Figure 9 sensors-19-03918-f009:**
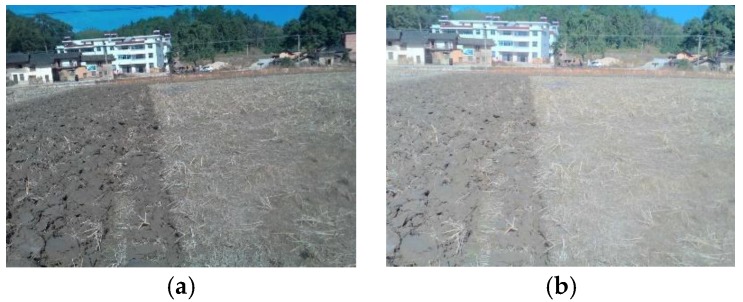
Fields with different illumination strength: (**a**) weak light intensity; (**b**) strong light intensity.

**Figure 10 sensors-19-03918-f010:**
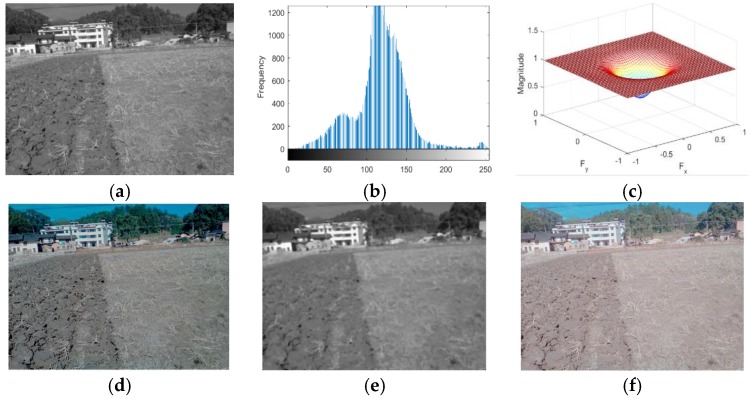
The process of the weak-light-intensity image using the homomorphic filtering (HF) algorithm: (**a**) grayscale; (**b**) grayscale histogram; (**c**) frequency response of the high-pass filter; (**d**) Butterworth high-pass filtering; (**e**) median filtering; (**f**) homomorphic filtering.

**Figure 11 sensors-19-03918-f011:**
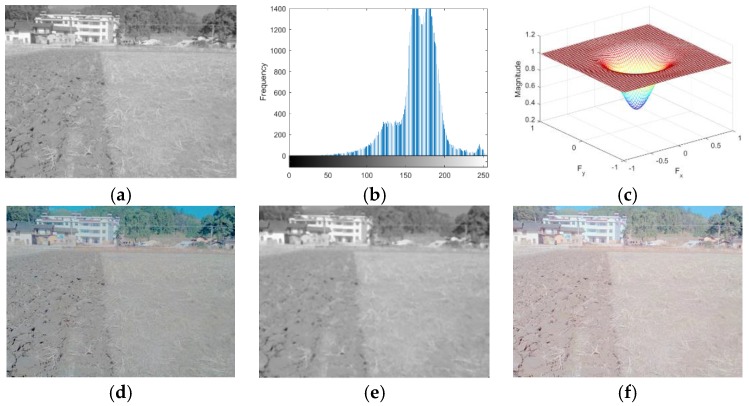
The process of the strong-light-intensity image using the homomorphic filtering algorithm: (**a**) grayscale; (**b**) grayscale histogram; (**c**) frequency response of the high-pass filter; (**d**) Butterworth high-pass filtering; (**e**) median filtering; (**f**) homomorphic filtering.

**Figure 12 sensors-19-03918-f012:**
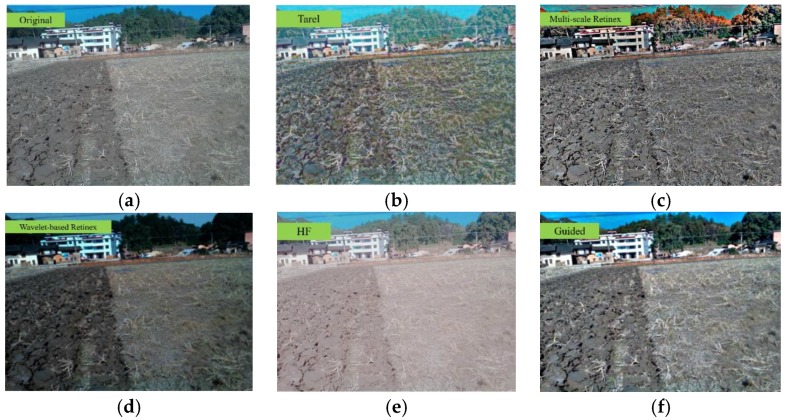
The input image and experiment results of the filtering algorithms: (**a**) original; (**b**) Tarel; (**c**) multi-scale retinex; (**d**) wavelet-based retinex; (**e**) HF; (**f**) guided.

**Figure 13 sensors-19-03918-f013:**
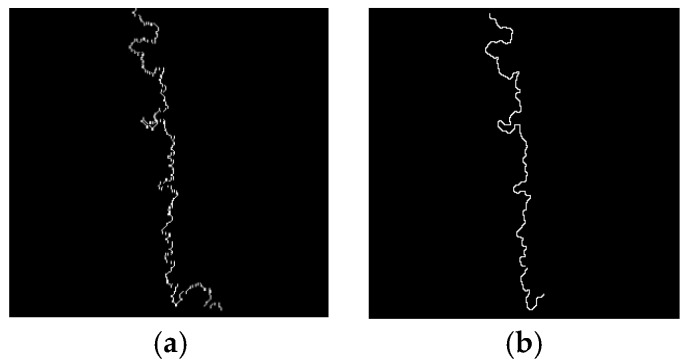
Edge results comparing different morphological processing: (**a**) basic morphology; (**b**) improved anti-noise morphology.

**Figure 14 sensors-19-03918-f014:**
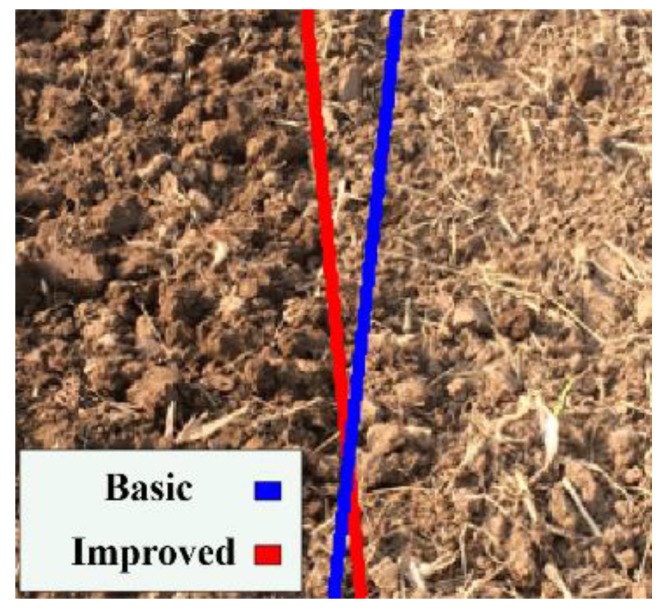
Results of Hough transform.

**Figure 15 sensors-19-03918-f015:**
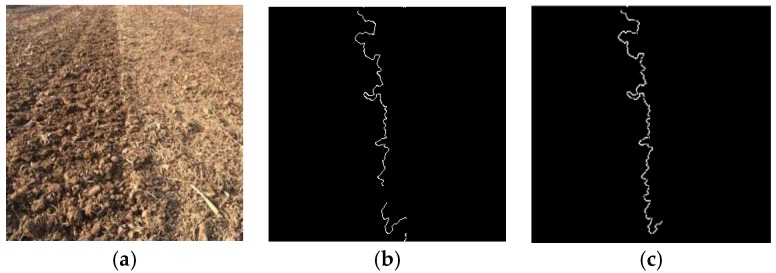
The results of edge detection: (**a**) original; (**b**) Sobel; (**c**) Roberts; (**d**) Prewitt; (**e**) Log; (**f**) improved anti-noise morphology.

**Figure 16 sensors-19-03918-f016:**
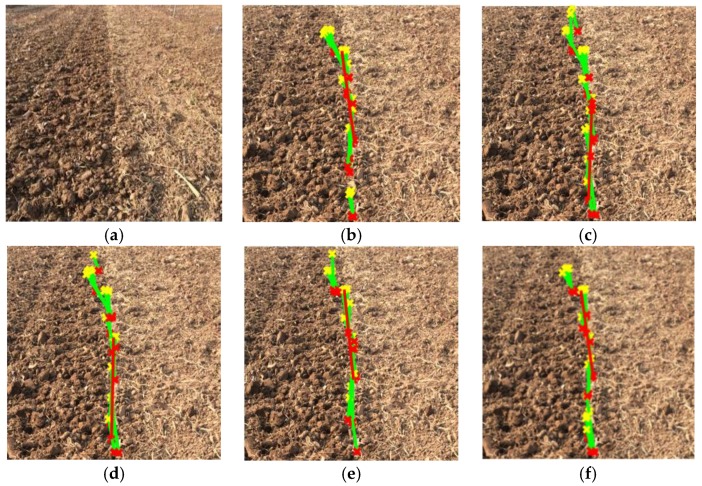
The longest line of the operators: (**a**) original; (**b**) Sobel; (**c**) Roberts; (**d**) Prewitt; (**e**) Log; (**f**) improved anti-noise morphology.

**Figure 17 sensors-19-03918-f017:**
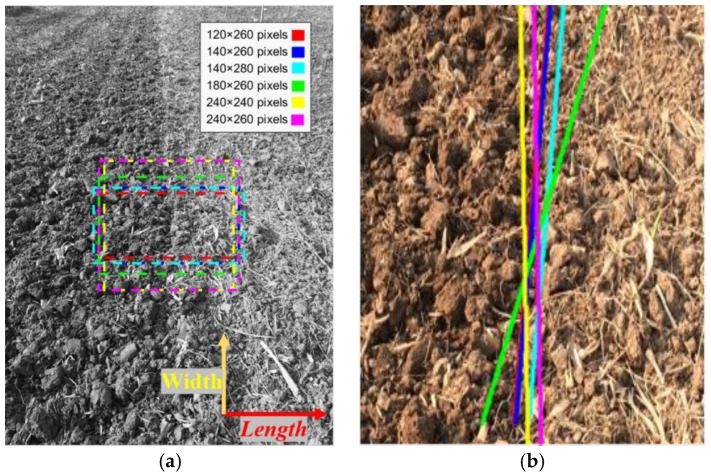
Different templates: (**a**) rectangular frame selection; (**b**) navigation line extraction result.

**Figure 18 sensors-19-03918-f018:**
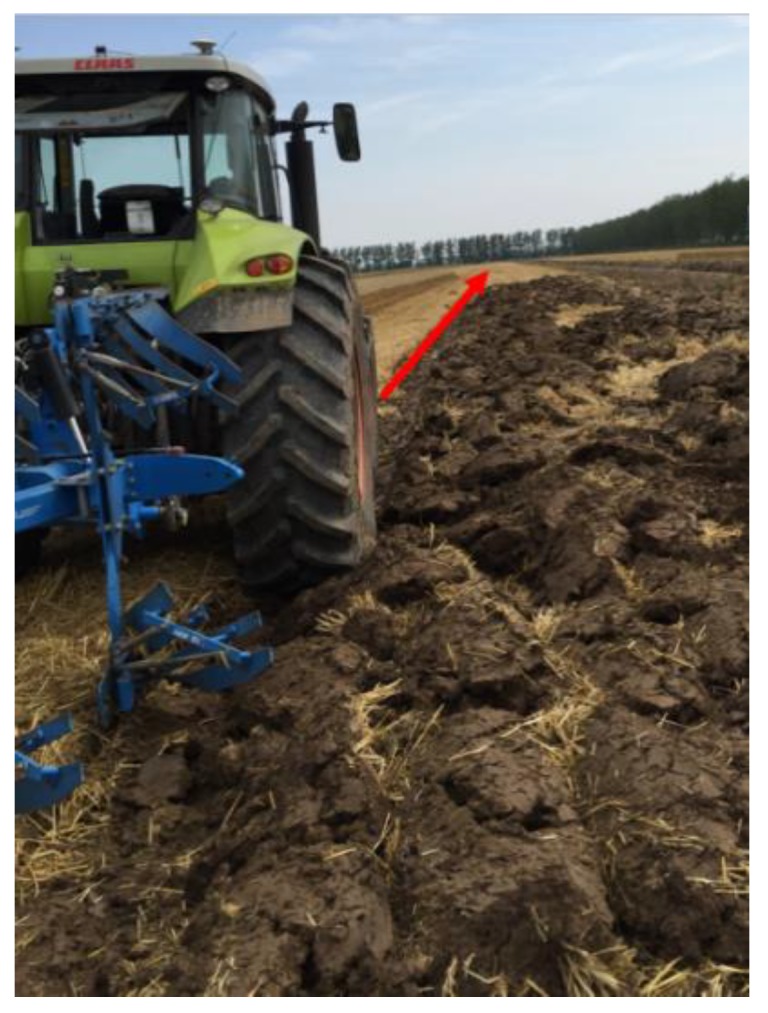
Tractor experiment diagram.

**Figure 19 sensors-19-03918-f019:**
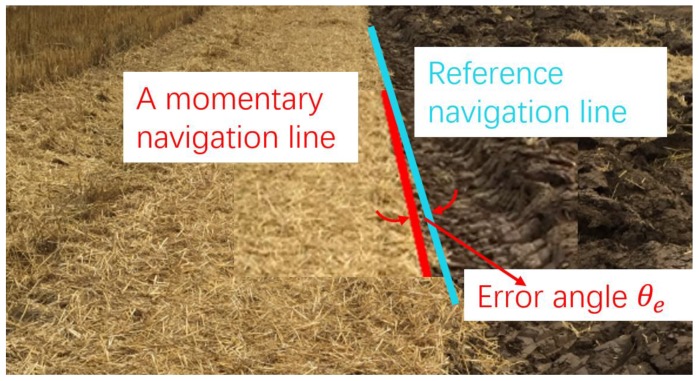
Navigation line extraction.

**Figure 20 sensors-19-03918-f020:**
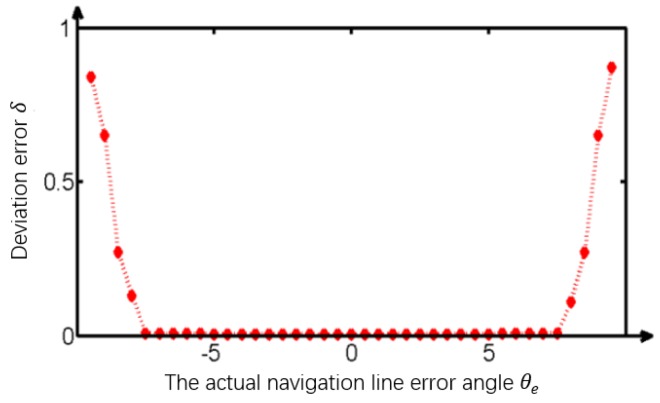
Error research.

**Figure 21 sensors-19-03918-f021:**
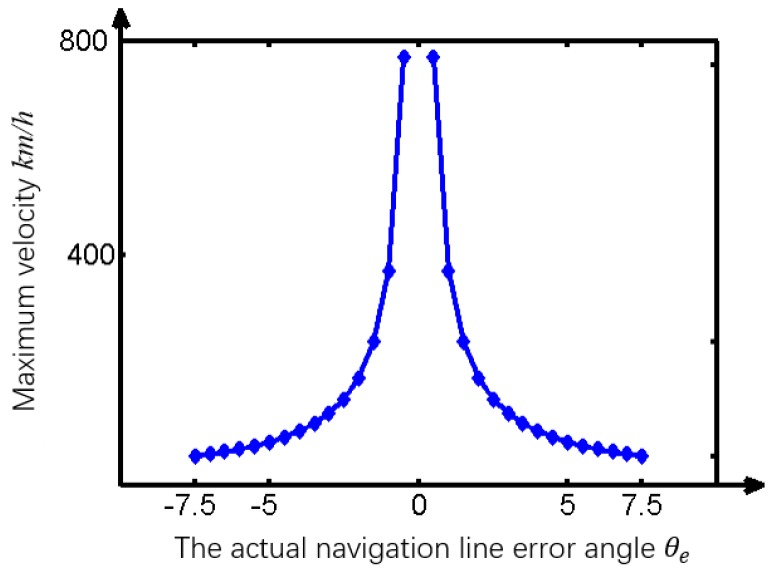
Maximum allowable velocity.

**Figure 22 sensors-19-03918-f022:**
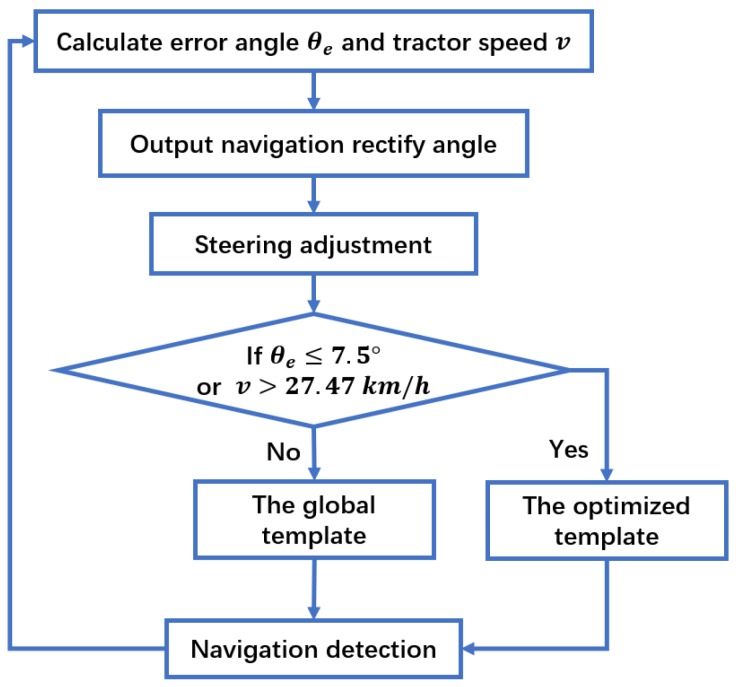
Template selection algorithm.

**Figure 23 sensors-19-03918-f023:**
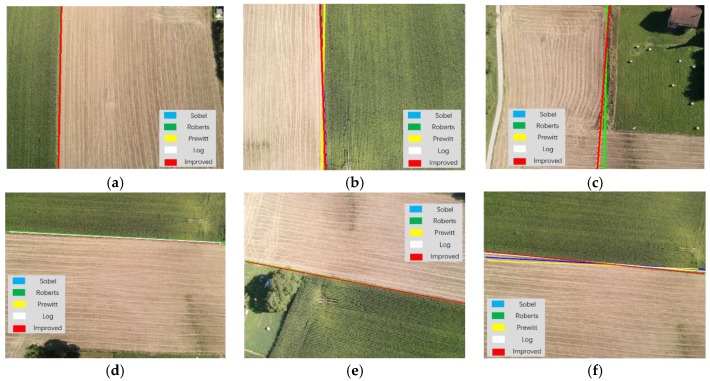
Comparison between the improved anti-noise morphology algorithm and existing operators using standard datasets. Red navigation lines were created using the improved anti-noise morphology algorithm: (**a**) first image sample; (**b**) second image sample; (**c**) third picture sample; (**d**) fourth picture sample; (**e**) fifth picture sample; (**f**) sixth picture sample.

**Table 1 sensors-19-03918-t001:** Consumption contrast of different color spaces.

Color Space	Time Consumption (s)
YCbCr	0.094
HSV	1.541
HIS	1.639
RGB	0.126

**Table 2 sensors-19-03918-t002:** Testing data for the different filtering methods.

Filtering Method	Highlighting	Time Consumption (s)
Tarel	−	0.902
Multi-scale retinex	+	0.552
Wavelet-based retinex	+	1.008
Homomorphic filtering (HF)	−	0.867
Guided	+	0.113

**Table 3 sensors-19-03918-t003:** Time consumption contrast of different edge operators.

Edge Operators	Time Consumption (s)
Sobel	0.089
Roberts	0.090
Prewitt	0.090
Log	0.096
Improved anti-noise morphology	0.073

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
