# Peer review of "Navigation Algorithm Based on the Boundary Line of Tillage Soil Combined with Guided Filtering and Improved Anti-Noise Morphology"

_sensors, 2019, doi:10.3390/s19183918_

Round 1

Reviewer 1 Report

Authors improved the MS accordingly with all the reviewers' suggestions. The paper could be accepted as it is.

Author Response

       Thanks a lot for your support and affirmation to our manuscript!

       We have revised the paper according to the suggestions of all the reviewers.

Best regards!

                                                      -Wei Lu

                                                       09/05/2019

Reviewer 2 Report

The authors have improved the overall quality of the paper, but has not resolved one of the main questions o the paper: how does their solution compared to other work?

There is no standard image set evaluation neither any comparison to other kind of image processing algorithms.

Test should be run on data set available here: https://www.sensefly.com/education/datasets/ or https://plantvillage.psu.edu/

The modifications to the tractor made is not detailed enough:

Where is the camera located? What processing capacity is on the tractor?

Author Response

       Thanks a lot for your valuable and professional suggestions on our manuscript!

 The replies to the comments are shown as follows.

To the comments of the 2nd reviewer:

Q1: how does their solution compared to other work?

A: Thanks for the good suggestion! Actually, there is no existing method for agricultural navigation using the new and old soil boundary till now in the literatures. According to the reviewer’s suggestion, we have downloaded 2 datasets from the website https://www.sensefly.com/education/datasets/ (we can’t find the similar images in the web https://plantvillage.psu.edu/ because most of them are about different species of fruit, flower, crop, etc.) and selected 8 images which are similar to the new and old soil boundary.  The effect comparation between the existing operators and the proposed improved anti-noise morphology was carried out, which is the main contribution in this paper, and the results show that the time consumption and navigation precision of our proposed algorithm have significant advantages in contrast with others as the whole. And the detailed processed results please find the attachment.

Table . Time consumption contrast of different edge operators using standard datasets.

Edge operators

Time consumption/s

Sobel

Roberts

Prewitt

Log

Improved anti-noise morphology

0.071

0.085

0.085

0.101

0.065

Q2: Where is the camera located? What processing capacity is on the tractor?

A: (1) The camera is mounted in the center of the front outside of the tractor as shown in Figure 1. (2) We employed the industrial computer for visual navigation using our proposed algorithm and the parameters of the industrial computer are as follows: 64-bit Windows 10, 8G system memory, Core(TM) i5 dual-core 2.80GHz. The navigation algorithm works good on the Matlab R2016a platform which can meet real-time visual navigation of a tractor in tillage mode.

We have revised all the above problems in the manuscript.

Thanks again for your valuable suggestions! It is of great significance for improving this article.

Best regards!

                                                        Wei Lu

                                                       9/4/2019

Reviewer 3 Report

I still have doubts about the terms "new/old" soil. Don't the authors really have an idea for a better terms?

Either way, the corrected manuscript presents a satisfactory level, although the language is worth refining.

Author Response

       Thanks a lot for your valuable and professional suggestions on our manuscript!

 The replies to the comments are shown as follows.

To the comments of the 3rd reviewer:

Q: I still have doubts about the terms "new/old" soil. Don't the authors really have an idea for a better terms?

A: We have discussed with other agricultural experts and reached a consensus that "new/old" soil may be replaced as “tillage soil boundary line” to reflect the academic meaning.

We have revised all the above problems in the manuscript.

Thanks again for your valuable suggestions! It is of great significance for improving this article.

Best regards!

                                                        Wei Lu

                                                       9/4/2019

Round 2

Reviewer 2 Report

The authors have replied to all comments/requests from the reviewers.

This manuscript is a resubmission of an earlier submission. The following is a list of the peer review reports and author responses from that submission.

Round 1

Reviewer 1 Report

Proposed manuscript contains quite interesting content, but its quality mitigates the enthusiasm. In my opinion, the Authors should rewrite it with an eye for a clear communication, fill in the gaps in the content (see comments below), perform additional experiments, and then its quality may turn out to be sufficient for publication.

1. Title should be changed, as it does not reflect the content of the work.

2. Nomenclature: "new/old soil", "traditional morphology", "advanced morphology", "structural elements"?

3. Organization of a manuscript. Section "Materials and methods" contains only some methods and no materials. Sections 3 and 4 should be part of it. The whole manuscript should be reorganized.

4. Description of "intelligent tractor" is not detailed enough. Also, where is the photo of "steering configuration" (Fig. 1 b)?

5. Language - should be improved (e.g. first sentences of Sections 2 and 4).

6. Language - not very scientific in certain places (e.g. "edge dealing"?).

7. Quality of Figs. 2, 5 and 22.

8. Section 4 - Structuring (not structural!) elements are given without honest explanation of their shape and size. In morphological image processing size of structuring element remains in close relation with resolution of processed image (or more precisely - the size / scale of the interesting elements in the picture).

9. Where are references for methods of image processing in Sections 3 and 4? Where is explanation for operators in Eqs. 7 and 8, and where is substantiation for these equations?

10. Fig. 9 - Comparison of different  views is pointless. It should be one view with different illumination.

11. Section 5.2 - how was the tractor guided? I presume that the described algorithms were implemented in some kind of microprocessor of CPLD/FPGA? It must be described in detail!

12. Where is discussion of results with literature?

Reviewer 2 Report

According to the authors, the paper describes an improved navigation, which is supposed to be working in agricultural environment.

The paper has two fields, which is dealing with: one part focuses on the vision algorithms, the other part focuses on the driving of the tractor. These two parts are heavily depending on each other and cannot be treated separately, like in the paper is presented today. The tractor cannot plow with 50 km/h, even the algorithm can provide such high frequency data. Not to mention the dynamics of the tractor and the plow operation.

The authors should separate the vision algorithms from the experiments with driving the tractor.

The results from the vision side, should be compared to realistic datasets available, like:

https://www.nature.com/articles/s41438-019-0151-5 or https://www.sciencedirect.com/science/article/pii/S0168169917305689

The autonomous plowing should be compared to existing solutions:

https://ieeexplore.ieee.org/abstract/document/7194284

Reviewer 3 Report

The paper is vary intersting and could return a great interest from the readers. I noticed some problems in the organization of the paper and a complete lack of the discussion section. These are my comments:
Sections 3-4 is a subsection of section 2 (M&M) and should be modified. Moreover some subsections of section 5 reported resuls and shoud be positioned in a different part of the paper.
A discussion section in light of the scientific literature is completely lacking and should be improved.
In order to further increase the efficiency of the proposes algorithm, do the authors cosider methods for color standardization, such as the one proposed by Menesatti et al. 2012 (3D Thin-Plate Spline) published on SENSORS? Please discuss.
Please substitute & with 'and' allover the text.